# Policy optimization in reinforcement learning for column generation

## Abstract

Column generation (CG) is essential for addressing large-scale linear integer programming problems in many industrial domains. While its importance is evident, the CG algorithms face convergence issues, and several heuristic algorithms have been developed to address these challenges. However, few machine learning and reinforcement learning methods are available that enhance the existing CG algorithm. This paper introduces a new policy optimization RL framework to improve the existing DQN-based CG framework, particularly training time, called **PPO-CG**. When applied to the Cutting Stock Problems (CSP), our approach requires merely **20%** of the training time observed with the DQN-based method and only **35%** in Vehicle Routing Problems with Time Windows (VRPTW). Our code is available in this link[1].

## 1 Introduction

Combinatorial optimization problems are widely applied in various industrial domains such as logistics, telecommunications, and transportation. Solving large-scale optimization problems efficiently is crucial for practical applications. In integer linear programming (ILP), the column generation (CG) technique is commonly used to solve large-scale integer programming problems.

The CG algorithms take advantage of the fact that the optimal solution only needs a part of the entire column, making it inefficient to consider the entire matrix. To begin with, the CG algorithm selects a subset of columns from the Master Problem (MP) and solves the relaxed linear programming problem for the selected columns, called the Restricted Master Problem (RMP). Then, using the dual variable, the CG algorithm solves the Pricing Problem (PP). The solutions of PP are new columns, which have the potential to improve the objective function. The procedure continues until there are no more columns to add.

Despite its usefulness, it is well-known that CG algorithms have convergence issues. Several heuristic algorithms have been developed to address these challenges. For a comprehensive overview, please refer to Lübbecke & Desrosiers (2005); Vanderbeck (2005) and the references within.

With the advancement of machine learning (ML) and reinforcement learning (RL), researchers are increasingly interested in solving combinatorial optimization problems and enhancing existing heuristic algorithms. For a review of this research domain, we direct readers to Mazyavkina et al. (2021); Bengio et al. (2021); Cappart et al. (2023). Moreover, recent work by Berto et al. (2023) presents a unified RL framework for combinatorial optimization problems.

In line with these growing research interests, applying ML or RL to improve existing heuristic algorithms is gaining significant attention. In Khalil et al. (2017), the authors leveraged supervised learning to enhance the branch-and-bound heuristic. The research presented in Tang et al. (2020) employed RL to refine the cutting plane method, yielding performance surpassing that of human-engineered heuristic algorithms. Meanwhile, Wu et al. (2021) introduced a customized actor-critic approach to improve existing large neighborhood search algorithms.

---

[1]https://anonymous.4open.science/r/PPO-CG/README.md

Compared with other heuristic algorithms, very little literature exists on CG algorithms employing ML or RL. To the best of our knowledge, Morabit et al. (2021) is an early attempt to use ML to improve CG. The authors generate multiple columns in each iteration and employ an "expert" system —represented as a mixed integer linear programming (MILP)— to supervise training column selection of the neural network. Datasets are collected from the "expert" before this training process. The authors then encode the RMP into bipartite graphs with column and constant nodes, as introduced by Gasse et al. (2019). The neural network is trained in a supervised manner to mimic the behavior of the "expert." A limitation of this approach is that solving time-consuming MILP problems is essential in the data collection phase.

The RL approach to CG is first proposed by Chi et al. (2022), called **RLCG**. In Chi et al. (2022), the authors follow the methodology from Morabit et al. (2021), employing the Deep Q-Network (DQN) algorithm for node selection. Additionally, they use a GNN as an approximator for the Q-function of the encoded RMP.

We are interested in improving the existing DQN-based approach using different RL algorithms, such as policy-based algorithms. This work provides an improved version of the RL framework for CG utilizing Proximal Policy Optimization (PPO), called **PPO-CG**. In the process, we combine actor and critic networks with GNN.

We conduct our experiments on two main tasks in the CG algorithms: the Cutting Stock Problems (CSP) and the Vehicle Routing Problems with Time Windows (VRPTW). Our approach, **PPO-CG**, requires only 20% of the training time observed in **RLCG** for the CSP task and 35% for the VRPTW task. Moreover, the experiment shows that **PPO-CG** are more robust than **RLCG**.

In summary, this paper offers the following contributions:

- Introduces a novel RL framework for column generation that, compared to the DQN-based approach, **RLCG**, achieves comparable performance and significantly reduces training time for both CSP and VRPTW tasks.

- Proposes a new method to integrate GNN with the actor-critic network during CG iterations.

## 2 Related Works

### 2.1 Basic Column Generation

In this subsection, we begin with the CG method for linear programming (LP) and then discuss how to get the integer solutions from the LP solution. Let us consider the following MP:

$$\min_x c^T x$$
$$\text{s.t. } Ax \leq b, \quad x \geq 0, \tag{MP}$$

where the matrix $A \in \mathbb{R}^{n \times m}$, vectors $x, c \in \mathbb{R}^m$ and $b \in \mathbb{R}^n$. If the number of columns $m$ is very large compared to $n$, we consider the following RMP

$$\min_{x'} (c')^T x$$
$$\text{s.t. } A'x' \leq b, \quad x' \geq 0, \tag{RMP}$$

where $A' \in \mathbb{R}^{n \times m'}$, vectors $x', c' \in \mathbb{R}^{m'}$ with $1 \leq m' \leq m$. Here, The columns of $A'$ are a subset of the columns of $A$. Denoting a dual variable of Equation (RMP) by $\lambda \in \mathbb{R}^n$, we solve the following PP:

$$\delta_i = c'_i - \sum_j A_{ji} \lambda_j. \tag{PP}$$

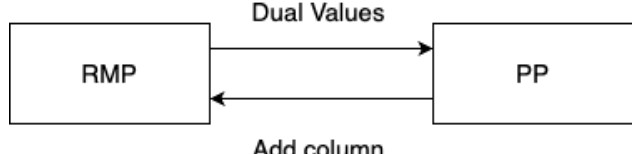

Figure 1: Oveall column generation process

If there exists $i$ such that a reduced cost $\delta_i < 0$, then we add $i$-th column of $A$ to $A'$ and iterate this process until no more column is selected (see Figure 1).

In this work, we address two ILP tasks: CSP and VRPTW. The CG process for ILP is similar to that used in LP, with only minor modifications. The simplest approach is to round LP solutions up to the nearest integer. For additional methods, please refer to Appendix C. For details about the ILP formulations of CSP and VRPTW, refer to Appendix A. Further comments on the CG iteration for these two tasks can be found in Appendix B.

## 2.2 Proximal Policy Optimization

The PPO algorithm was introduced in the seminal work by Schulman et al. (2017) as an easy-to-implement alternative to Trust Region Policy Optimization (TRPO) by Schulman et al. (2015). Recently, a theoretical study on the convergence of the PPO algorithm is studied in Holzleitner et al. (2021), and a general theoretical framework that includes PPO is introduced by Fan & Xiao (2022). In this work, we apply PPO to the CG iteration, and the motivation for choosing PPO is discussed in Section 3.2.

Let us now briefly explain the PPO algorithm. Detailed information and related topics can be found in Achiam (2018). Let us denote $\mathcal{S}$ and $\mathcal{A}$ as a state and action space respectively. For an action $a \in \mathcal{A}$ and a state $s \in \mathcal{S}$, we denote a parameterized policy by $\pi_\theta = \pi_\theta(a|s)$. Denoting $Q^{\pi_\theta}$ for the on-policy action-value function, the policy value function $V^{\pi_\theta}$ and advantage function $A^{\pi_\theta}$ are respectively defined as

$$V^{\pi_\theta}(s) = \mathbb{E}_{a \sim \pi_\theta}[Q^{\pi_\theta}(s,a)] \quad \text{and} \quad A^{\pi_\theta}(s,a) := Q^{\pi_\theta}(s,a) - V^{\pi_\theta}(s).$$

In addition, we define the ratio function and clip function as:

$$r(\theta, \theta_{\text{old}}, s, a) := \frac{\pi_\theta(a|s)}{\pi_{\theta_{\text{old}}}(a|s)} \quad \text{and} \quad \text{clip}(t, t_{\min}, t_{\max}) := \max(t_{\min}, \min(t, t_{\max})),$$

for $t, t_{\min}, t_{\max} \in \mathbb{R}$. Then the PPO algorithm updates a parameterized policy $\pi_\theta$ by maximizing:

$$L^{clip}(\theta, \theta_{\text{old}}) = \mathbb{E}_{s, a \sim \pi_{\theta_{old}}} \left[ \min(r(\theta) A^{\pi_{\theta_{\text{old}}}}(s,a), \text{clip}(r(\theta), 1 - \varepsilon, 1 + \varepsilon) A^{\pi_{\theta_{old}}}(s,a)) \right]. \quad (1)$$

Note that we simplify $r(\theta)$ to denote the ratio function, and $\varepsilon > 0$ is a hyperparameter to be determined later in Section 4.

## 3 Proposed Methods

In this section, we provide our proposed method, which is motivated by the methods provided in Morabit et al. (2021) and Chi et al. (2022). The DQN-based approach in Chi et al. (2022) is called **RLCG**, and we name our method as **PPO-CG**.

### 3.1 Markov decision process formulation

We define the transition map $\mathcal{T} : \mathcal{S} \times \mathcal{S} \times \mathcal{A} \to \mathbb{R}$ as $T(s, s', a) = P(s'|s, a)$, denote the reward function by $reward : \mathcal{S} \times \mathcal{A} \to \mathbb{R}$, and represent the discount factor by $\gamma \in [0, 1]$.

**State $\mathcal{S}$**

For each iteration of the CG, we represent the matrix in (RMP) as a bipartite graph composed of column nodes $\mathcal{X}$ and constraint nodes $\mathcal{B}$, as in Gasse et al. (2019). There exists an edge connecting $(x, b) \in \mathcal{X} \times \mathcal{B}$ if the column contributes to the constraint $c$ (see Figure 2). We set the node features depending on the task as specified in Appendix L.

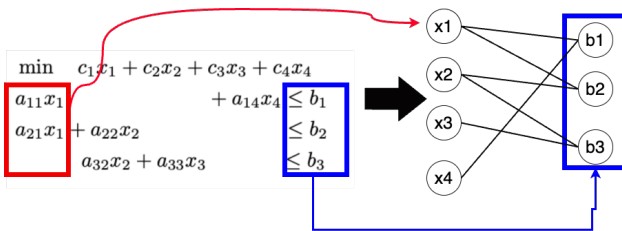

Figure 2: Each column is represented as the node in column nodes in $\mathcal{X}$, and constraints are represented as constraint node class $\mathcal{B}$. For instance, if a column $x_1$ contributes to the constraints $b_1$, an edge exists between $x_1$ and $b_1$.

**Action $\mathcal{A}$ and Transition $\mathcal{T}$**

By solving Equation (PP), we find candidate columns with negative reduced cost. The action is to choose the next column or node to add to the current RMP or graph (see Figure 3). The maximum number of candidates at each iteration is also a hyperparameter.

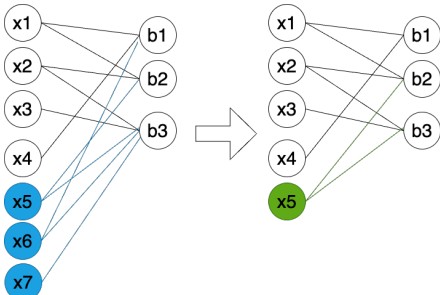

Figure 3: In this example, after PP is solved, there exist three candidates (blue nodes), then choose one node (green node).

**Reward**

Since our purpose is to achieve a lower objective function, —we are solving minimizing problem— within fewer iterations, we set the reward as

$$reward_t = \alpha \left( \frac{obj_{t-1} - obj_t}{obj_0} \right) - p,$$

where $\alpha > 0$ and $p > 0$ are hyperparameters. The parameter $p > 0$ is the iteration penalty, giving a negative reward if the model can not finish iteration after the action is chosen.

## 3.2   PPO-CG

This subsection introduces the architecture and overall framework of **PPO-CG**. We begin by explaining why the PPO algorithm was chosen in the context of CG.

**Motivations**

Let us revisit the DQN algorithm in the CG setting. Note that in the context of CG, the order in which columns are chosen is not important. Suppose that two trajectories, $\tau_1 = [s_0, a_0, s_0 + a_0, a_1, s_0 + a_0 + a_1]$ and $\tau_2 = [s_0, a_1, s_0 + a_1, a_0, s_0 + a_0 + a_1]$, are given. Each action $a_i$ implies the choice of a column in matrix $A$, and the addition notation denotes the inclusion of the column. If there exist parameters $\theta_1$ and $\theta_2$ satisfying

$$a_1 = \arg\max_a Q_{\theta_1}(s_0, a) \quad \text{and} \quad a_2 = \arg\max_a Q_{\theta_1}(s_0 + a_1, a),$$

$$a_2 = \arg\max_a Q_{\theta_2}(s_0, a) \quad \text{and} \quad a_1 = \arg\max_a Q_{\theta_2}(s_0 + a_2, a),$$

then, in the CG iteration, the difference between using $Q_{\theta_1}$ and $Q_{\theta_2}$ is expected to be very small. The final objective values are the same, and there might be very few differences in the number of iterations and execution times because solving PP mostly depends on the problem size and the number of selected columns, not on the specific choice of the column. We believe this phenomenon can lead to inefficiency in training and require more resources because we need to update $Q_\theta$ more frequently, particularly when the number of actions and states is very large, as in CG.

Furthermore, selecting as few columns as possible is important, so in the training process, we want to use the information in the trajectory instead of using the replay buffer. To capture these properties of CG, we select an online, model-free RL algorithm that does not need a replay buffer and is a policy-based RL algorithm that has both critic and actor networks —based on a previous study (Chi et al., 2022) that shows DQN can improve traditional CG algorithms— and is easy to implement. The algorithm chosen for this study is the PPO algorithm, which meets these criteria. Moreover, the PPO algorithm has theoretical support. See, for instance, (Holzleitner et al., 2021; Fan & Xiao, 2022).

**Model architecture**

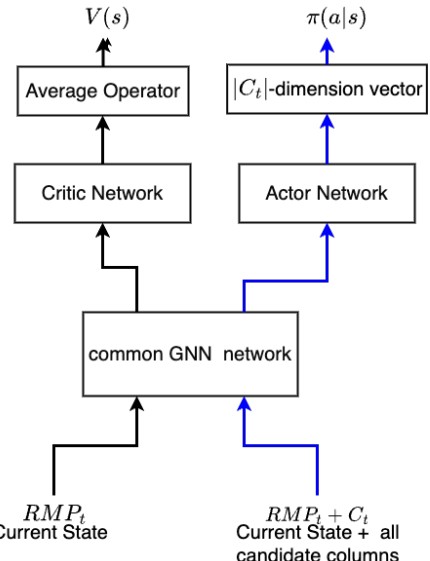

Figure 4: actor and critic networks with input $RMP_t$ and $RMP_t + C_t$.

We employ the GNN layers introduced in Morabit et al. (2021) with the actor-network and the critic-network sharing common layers. We denote $t \in \mathbb{N} \cup \{0\}$ for the CG iteration step. Let $RMP_t$ denote Equation (RMP) or the corresponding graph. We use $RMP_t + C_t$ to represent the graph of Equation (RMP), including all candidate nodes. Moreover, the notation $|C_t|$ implies the number of candidate nodes. For each $t$, solving Equation (PP) gives $|C_t|$ candidate nodes to choose from. Then, we take $RMP_t$ and $RMP_t + C_t$ as inputs of **PPO-CG**. First, $RMP_t$ passes through a common layer, followed by the critic-network. Next, $RMP_t + C_t$ passes through the common layer and the actor-network.

Both the actor and critic networks have the same structures. Both networks return $|C_t|$-dimension vectors. We get the value function $V(s)$ by taking an average of $|C_t|$-dimension vectors. We get a policy $\pi(a|s)$ from the output of the actor-network. Since $RMP_t$ and $RMP_t + C_t$ have similar graph structures except for the number of nodes, using a common layer for the actor and critic network seemed natural. See Figure 4 for the overview of the proposed architecture.

**Overall framework**

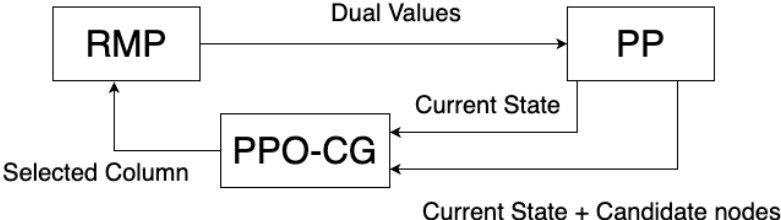

Figure 5: **PPO-CG** framework.

From the model architecture depicted in Figure 4, we now provide an overall framework of **PPO-CG**, summarized in Figure 5. Algorithm 1 describes the training procedure of **PPO-CG**. Once training **PPO-CG** is done, the model only uses the actor-network to choose the next column in the CG iteration in Figure 5. We denote the GNN model described in Figure 4 with parameter $\theta$ as **PPO-CG**$_\theta$, and $RMP_t + a_t$ for

---

**Algorithm 1** Triaining procedure

Initialize **PPO-CG**$_{\theta_{\text{old}}}$ and **PPO-CG**$_\theta$
**for** problem p = 1, 2, $\cdots$ **do**
    $RMP_0$=Initialize($p$) ; $t = 0$;
    **for** epoch = 1 to $E$ **do**
        **while** True **do**
            Forumate Equation (PP) via duality of $RMP_t$
            Get $C_t$ by solving Equation (PP)
            **if** $|C_t| = 0$ **then**
                Break;
            **end if**
            Calculate: advantage function $A_t$, action $a_t$ from **PPO-CG**$_{\theta_{\text{old}}}(RMP_t, RMP_t + C_t)$
            Update: $RMP_{t+1} \leftarrow RMP_t + a_t$, $t \leftarrow t + 1$,
        **end while**
        Compute $L^{CLIP}(\theta)$ using Equation (1).
        Update $\theta$ by maximizing $L^{\text{CLIP}}(\theta)$.
        Update $\theta_{\text{old}} \leftarrow \theta$
    **end for**
**end for**

---

adding the next column $a_t$ to the current state $RMP_t$.

### 3.3 Comparision with RLCG

Since our architecture is based on **RLCG**, it would be great to point out differences made on **PPO-CG**. In **RLCG**, only the actor network is used. Also, it takes only $RMP_t + C_t$ as input only uses the actor-network. The output of the actor-network is considered as an action-value function, $Q(s, a)$. Whereas we take both $RMP_t$ and $RMP_t + C_t$, and the output of the actor is considered as the policy $\pi(a|s)$ and the average of the output of critic network is considered as the value function $V(s)$.

## 4    Experiments

In this section, we outline the details of our experimental process. We use TensorFlow 2.13 and the free version of GurobiPy 10.0.3. **RLCG** is the baseline method for comparison and the official implementation available.[2] For the hyperparameters used in training **RLCG**, please refer to Appendix D.

### Summary of training process

Since our main contribution is to reduce the training time, we first summarised the time it takes to train each model in Table 1. More information, such as statistical data and the time consumed for each model to

Table 1: Overall training time for each task and algorithm

| Tasks | Methods | training time(hours) |
|-------|---------|----------------------|
| CSP | **RLCG** | 319.9 |
| | **PPO-CG** (ours) | **66.8** |
| VRPTW | **RLCG** | 47.9 |
| | **PPO-CG** (ours) | **16.7** |

train each instance, can be found in Appendix F. The main reason for the reduction is that, as mentioned in Section 3.2, the PPO algorithm does not update parameters until each training instance is solved, whereas in DQN, parameters are updated every time an action is chosen. An additional experiment is conducted in Appendix H by adjusting the parameter update frequency in **RLCG** to provide more insight and data.

### 4.1    CSP tasks

### Dataset and machine specification

We train **PPO-CG** and **RLCG** using BPPLIB from (Delorme et al., 2018). In the training process, we use ILP instances with roll length sizes of 50, 100, and 200. The total number of instances is 439. In the test process, we use ILP instances with roll length sizes of 200 and 750 with 86 and 21 instances, respectively. We use NVIDIA RTX A5000 24GB GPU with Intel(R) Core(TM) i9-10980XE CPU @ 3.00GHz for this task. We refer to Appendix E for additional information on train and test dataset.

### Hyperparameters & node features

We set the learning rate for both the critic-network and actor-network to be $1e^{-4}$, the hidden dimension of the GNN model is 32, step penalty of $p = 10$, epoch size $E = 20$, $\varepsilon = 1e^{-2}$, action candidate size $= |C_t| = 10$. We set objective hyperparameter $\alpha = 100$ and reward decay exponent $\gamma = 0.999$. The variable nodes that belong to $\mathcal{X}$ have 9 node features and constraint nodes that belong to $\mathcal{B}$ have 2 node features. We refer to the details of node features in Appendix L.

### Test results

We compare test results from our model **PPO-CG**, DQN-based model **RLCG**, Greedy method, and expert methods. We compare each method using three metrics: objective function, time to execute, and number of iterations. First, we provide test results on roll length $n = 200$ and $n = 750$, respectively. Figure 6 and Figure 7 provide comparison results between **PPO-CG**. In Figure 8, we provide a boxplot for $n = 200$ and $n = 750$ with two metrics: execution time and number of iterations.

Based on the first column of Figure 6 and Figure 7, there is little difference in the objective value function across sizes and methods. Therefore, important metrics are execution time and iteration number. As depicted

---

[2]https://github.com/khalil-research/RLCG/

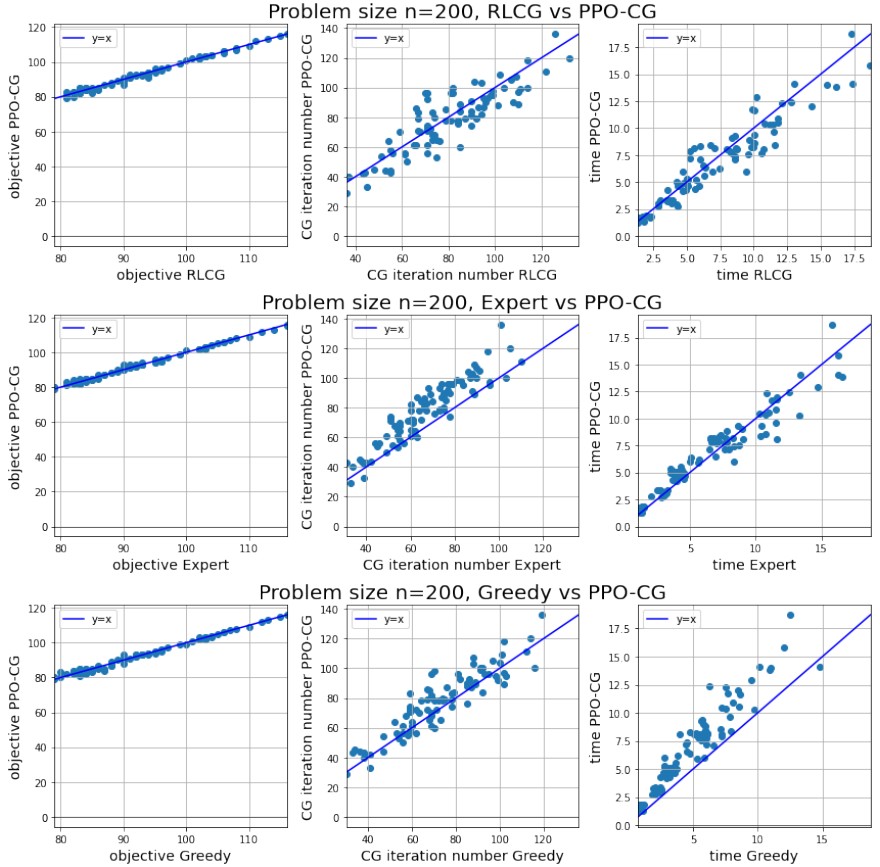

Figure 6: This plot displays the testing results on problem size $n = 200$ in CSP, with each of the 86 dots representing the outcome of a test instance. The first column compares the objective value of **PPO-CG** against other methods (**RLCG**, Expert, Greedy). Most points lie on the $x = y$ line, indicating similar objective values across different methods. The second column compares the number of CG iterations between **PPO-CG** and other methods. Points below the $y = x$ line signify that **PPO-CG** requires fewer iterations. The third column shows the runtime measured in seconds for each test instance against **PPO-CG**. Points below the $y = x$ line indicate that **PPO-CG** solve the instance faster. Overall, the results indicate that **PPO-CG** solves ILP problems faster than Expert but does not outperform the other methods for CSP problems of size $n = 200$.

in Figure 8, the Greedy and Expert methods are more effective when $n = 200$ compared to when $n = 750$. Moreover, out of 86 test instances of size 200, the Greedy algorithm outperforms **RLCG** on 84 instances and outperforms **PPO-CG** on 85 instances. Expert outperforms **RLCG** on 61 instances and outperforms **PPO-CG** on 57 instances.

However, for larger problems, $n = 750$, **PPO-CG**, and **RLCG** show significantly better performance in execution time. On 21 instances **RLCG** outperforms the Greedy algorithm on 13 instances and the Expert algorithm on 20 instances. **PPO-CG** outperforms the Greedy algorithm on 15 instances and outperforms the Expert algorithm on all instances.

We believe the superior performance of the Greedy algorithm for smaller problem sizes can be attributed to its independence from the GPU. This independence eliminates data transitions between the CPU and GPU, resulting in increased efficiency. Meanwhile, the Expert method has the lowest iteration number, but as the problem size expands, its execution time grows, making it the slowest of the four methods. This increase is primarily because the Expert method requires solving MILP. For smaller problems, the increment in solving additional MILP reduces the iteration number, and the execution time increase is not as critical. However,

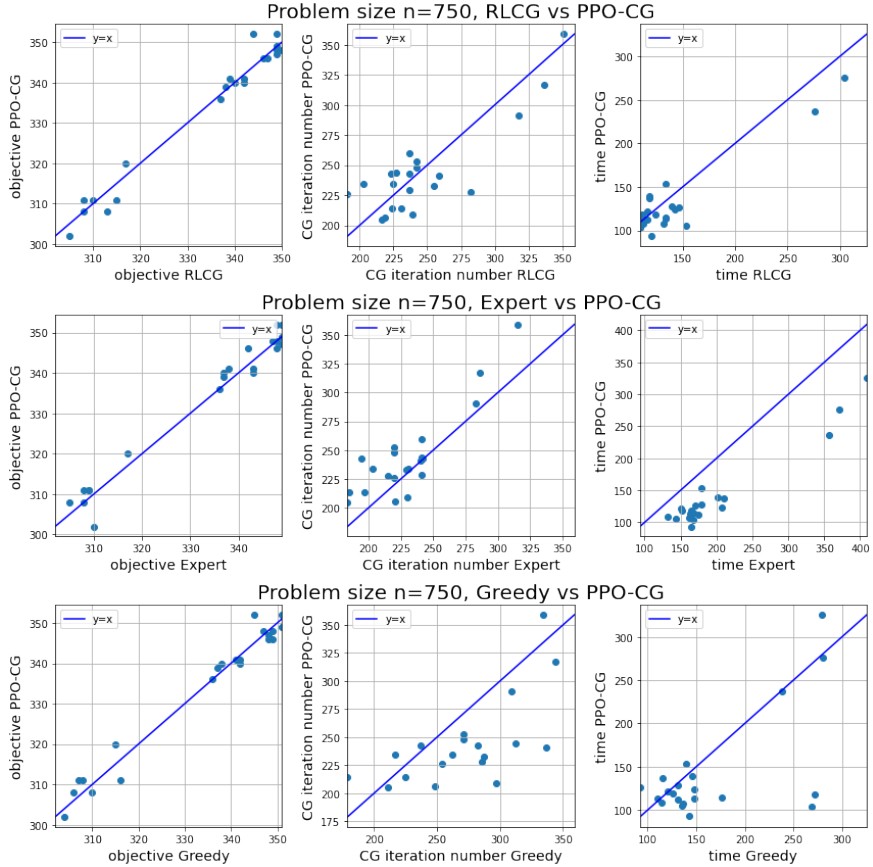

Figure 7: This plot displays the testing results for problem size $n = 750$ in CSP, with each of the 21 dots representing the outcome of a test instance. The first column compares the objective value of **PPO-CG** against other methods (**RLCG**, Expert, Greedy), showing that most points lie on the $x = y$ line, indicating similar objective values across methods. The second column compares the number of CG iterations, showing that points below the $y = x$ line imply that **PPO-CG** requires fewer iterations. The third column illustrates the runtime measured in seconds for each test instance, with points below the $y = x$ line indicating that **PPO-CG** solves test instance faster. Overall, **PPO-CG** solves test instances faster than **RLCG** on 15 out of 21 instances, faster than the Expert method for all test instances, and faster than the Greedy method on 15 instances, indicating that **PPO-CG** outperforms other methods in general.

this benefit diminishes for larger problems. Consequently, for these larger problems, the RL-guided method demonstrates notable improvements in execution time with smaller variances, even with data transition costs between CPU and GPU.

## 4.2 VRPTW tasks

**Dataset and Machine specification**

We train **PPO-CG** and **RLCG** using Solomon benchmark from (Solomon, 1987). In the training process, we use ILP instances with different types of problems and various numbers of customers. The total number of instances is 234, but 219 instances are used since the remaining 15 instances take too long to solve. We use ILP instances with 37 instances in the test process. We use NVIDIA GeForce RTX 4090 for this task with AMD Ryzen 9 7950X3D 16-Core Processor. We refer to Appendix E for additional information about the dataset.

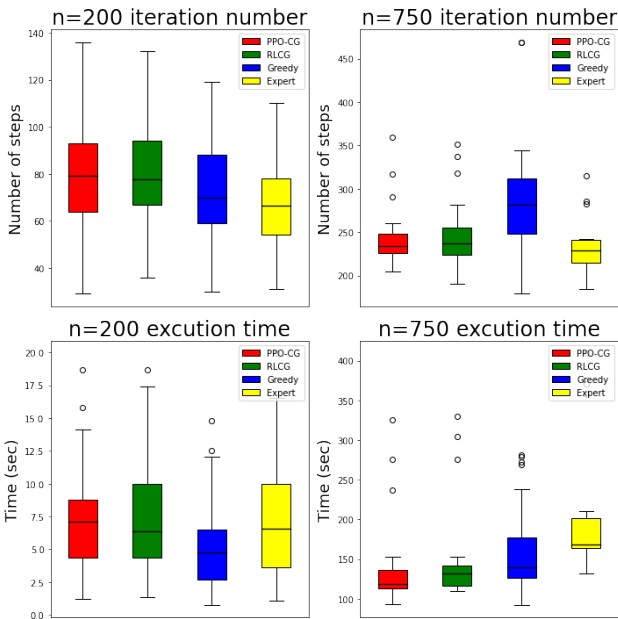

Figure 8: Boxplot comparing execution time and CG iteration numbers for four methods at $n = 200$ and $n = 750$ in CSP. At $n = 200$, **PPO-CG** and **RLCG** do not perform significantly better than heuristic methods. In contrast, at $n = 750$, **PPO-CG** and **RLCG** have lower median execution times than the other two heuristic methods. In addition, we claim that **PPO-CG** performs better than **RLCG** given that it has a lower median number for both metric and smaller interquartile ranges.

### Hyperparameters & node features

Many of the hyperparameter settings are similar to CSP. We set the learning rate for both the critic-network and actor-network to be $1e^{-4}$, the hidden dimension of the GNN model is 32, step penalty of $p = 10$, epoch size $E = 20$, $\varepsilon = 1e^{-2}$. Since the heuristic algorithm for solving Equation (PP) in VRPTW does not create a fixed number of columns, action candidate size $|C_t|$ is not fixed. We set objective hyperparameter $\alpha = 0.5$ as in the official implementation of **RLCG** and the reward decay exponent $\gamma = 0.999$. The variable nodes that belong to $\mathcal{X}$ have 8 node features, and constraint nodes that belong to $\mathcal{B}$ have 2 node features. We refer to Appendix L for details on node features.

### Remark: additional detail

In VRPTW, the heuristic algorithm for solving Equation (PP) takes too much time and does not generate a fixed number of candidate nodes. For these reasons, we use the $argmax$ argument to choose the next node as in **RLCG**, and instead of generating $E$ trajectories, we only generate one trajectory and update parameters $E$ times. This method has a shorter training time and shows almost similar performance results compared with creating $E$ numbers of trajectories. For details, we refer to Appendix G. From these observations, it seems possible to suggest a better direction than the current algorithm. It is likely to be an off-policy algorithm that utilizes previously used trajectories.

### Test results

We compare test results in our model **PPO-CG**, **RLCG**, and the Greedy method. As in the CSP task, we use three metrics to compare each method: objective function, execution time, and number of iterations. Figure 9 compares each method with **PPO-CG**. In Figure 9, we excluded an outlier during plotting to show the overall depiction clearly. The information about the excluded instance is summarized in Table 9. To enhance clarity, a boxplot is also presented in Figure 10. We claim that **PPO-CG** performs better than the other two models as problems get more complicated since average execution time and average iteration

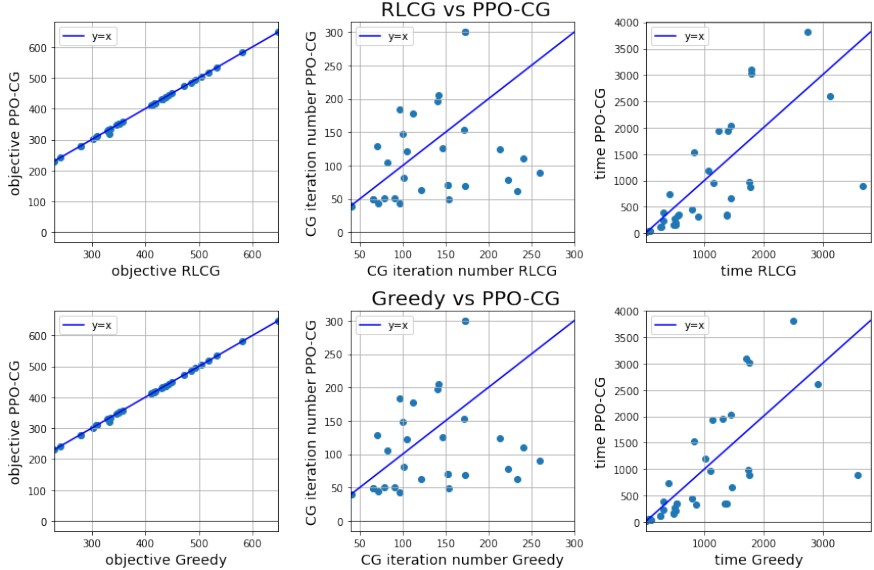

Figure 9: This plot displays the testing results for VRPTW, with each of the 36 dots representing the outcome of a test instance, excluding one extreme case described in Table 9. The first column compares the objective value of **PPO-CG** against other methods (**RLCG**, Greedy), showing that most points lie on the $x = y$ line, indicating similar objective values across methods. The second column compares the number of CG iterations, revealing that points below the $y = x$ line signify that **PPO-CG** requires fewer iterations. The third column illustrates the runtime measured in seconds for each test instance, with points below the $y = x$ line indicating that **PPO-CG** solve each test instance faster. Overall, **PPO-CG** runs faster than **RLCG** on 24 out of 36 instances, faster than the Greedy method on 24 instances, indicating that **PPO-CG** generally runs faster than other methods.

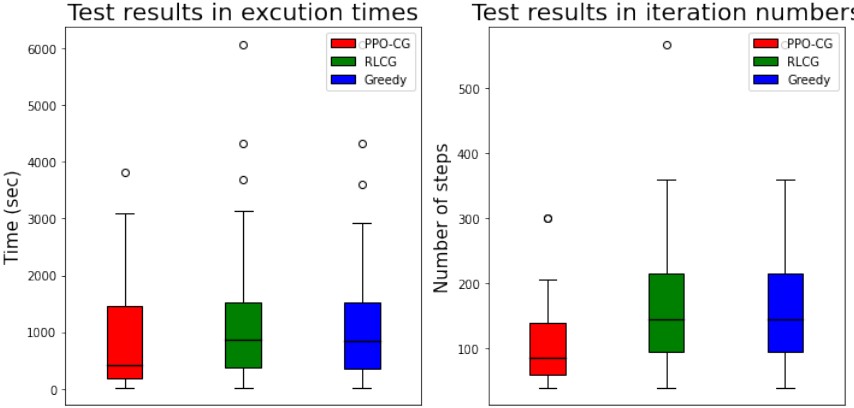

Figure 10: Boxplots compare three methods using two metrics: execution time and iteration numbers. For this task, **PPO-CG** generally performs better than the other methods in execution time and iteration numbers, as **PPO-CG** has the lowest median value for both metrics, but **PPO-CG** does not have the interquartile range for execution time.

numbers are lower. However, **PPO-CG** does not have significantly smaller interquartile ranges, indicating that our model, **PPO-CG**, might have a stability issue, which is a topic of future topic.

## 4.3 Convergence Analysis

Figure 11 displays the convergence analysis for CSP and VRPTW test instances. Objective values for each instance are normalized from 0 to 1, where 1 indicates optimal performance. Bold lines show the mean objective values across iterations, while shaded areas represent $\pm 1$ standard deviation. For CSP, we evaluate instances of sizes $n = 200$ and $n = 750$, showing similar convergence rates for **PPO-CG** and **RLCG**. In contrast, **PPO-CG** achieves faster convergence in VRPTW. We refer to Appendix M for details.

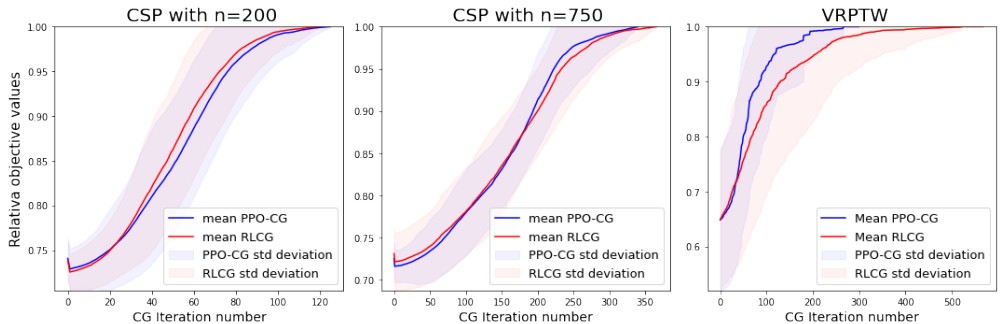

Figure 11: convergence graph on relative objective values for **PPO-CG** and **RLCG**

## 4.4 Remark on the robustness of RLCG and PPO-CG

We find that **RLCG** tends to behave like a Greedy algorithm in large-scale problems if hyperparameter-tuning is not properly done. For details, we refer to Appendix I. This suggests that although **RLCG** is an interesting approach, it might not be appropriate for large-scale CG problems.

# 5 Conclusions, limitations, and future works

This work proposes an improved RL framework for the CG algorithm. Our experiments, focused on two main tasks —CSP and VRPTW— empirically demonstrate that our model trains faster and exhibits comparable performance to the DQN-based model developed in Chi et al. (2022).

Our method still takes much time compared in training. The main reason for the time consumption is the high dependency of this method on solving Equation (PP) and moving data from CPU to GPU. If solving Equation (PP) takes a long time, then training the RL model also takes a long time, highlighting the need for further development. From the ablation study reported in Appendix H, for $n = 750$, solving PP and moving data to GPU takes 10% of total training time in the CSP problem. In VRPTW, some instances are removed from the train set because solving PP takes too long. Moreover, since PPO is on-policy learning, **PPO-CG** may suffer from sample efficiency, and as mentioned in Appendix G, it is observed that reusing already explored trajectories in training has the potential for reducing training time but still obtaining comparable model performance. For the future study, it would be interesting to compare other types of RL algorithms such as TD3 (Fujimoto et al., 2018), SAC (Haarnoja et al., 2018), IMPALA (Espeholt et al., 2018).

### Acknowledgements

We thank the reviewers for their attention and time spent reviewing this manuscript.

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

# A   Problem description and the ILP formulation of CPS and VRPTW

In this section, we shall provide details on the formulation of the CSP and VRPTW.

## CSP

The purpose of the CSP is to cut as few stocks as possible but satisfy the given demand $\{d_j\}_{j=1}^n$ for each item. Suppose that the given length of the stock is $L$, there are $n$ types of items, and for each item, $w_j$ length is required. The formulation of CSP follows from Delorme et al. (2018). We denote $u$ by the number of all the cutting patterns of the stock. For each item $j \in \{1, \cdots, n\}$, there are $d_j$ demands. Denoting $a_{ji}$ by the number of item $j$ in the $i$-th pattern. Let $x_i$ denote the decision variables such that $x_i = 1$ if $i$-th pattern is chosen, 0 otherwise. Then, we need to minimize the following integer linear programming:

$$\min_x \sum_{i=1}^{u} x_i$$

satisfying the following constraints:

$$\sum_{i=1}^{u} a_{ji} x_i = d_j, \quad \sum_{j=1}^{n} a_{ji} w_j \leq L, \quad x_i \in \{0,1\} \quad \text{and} \quad a_{ji} \geq 0, \text{ integer.}$$

## VRPTW

This section summarizes the context presented in Kallehauge et al. (2005). Let $\mathcal{C}$ be a set of customers and $\mathcal{G}$ be a directed graph with a number of nodes equal to $|\mathcal{C}| + 2$. We denote the set of vehicles by $\mathcal{V}$. Customers are represented by $1, 2, \cdots, n$, and each node in $\mathcal{G}$ is denoted by $0, 1, \cdots, n+1$. The routing of each vehicle starts at node 0 and ends at node $n + 1$. For simplicity, we use the notation $\mathcal{N} = 0, \cdots, n + 1$. For every node $i = 1, \cdots, n$, there is a corresponding customer $i$ with a time window $[a_i, b_i]$. Vehicles must arrive at node $i$ within the time window $[a_i, b_i]$. Also, we assume that $[a_0, b_0] = [a_{n+1}, b_{n+1}]$. The matrices $c_{ij}$ and $t_{ij}$ indicate the cost and time, respectively, that a vehicle takes by moving from node $i$ to node $j$. It is assumed that $c_{ij}$ and $t_{ij}$ satisfy the triangle inequality. Let $q$ denote the vehicle's capacity, and each customer $i$ has a demand $d_i$. We want each customer to be served exactly once and satisfy one's demand. To formulate ILP, let us introduce two decision variables, $x_{ijk}$ and $s_{ik}$. For each $(i,j) \in \mathcal{N} \times \mathcal{N}$ with $i \neq n+1, j \neq 0$ and $i \neq j$, we define

$$x_{ijk} = \begin{cases} 1 & \text{if vehicle } k \text{ move from vertex } i \text{ to vertex } j \text{ directly,} \\ 0 & \text{otherwise.} \end{cases}$$

The other decision variable $s_{ik}$ denotes the time vehicle $k$ starts to serve customer $i$. In case $i = 0$, we assume that $sik = 0$, since we assume that $a_0 = 0$. If the vehicle $k$ does not serve the customer $i$, it is an irrelevant variable. The ILP formulation for the VRPTW follows:

$$\min \sum_{k \in \mathcal{V}} \sum_{i \in \mathcal{N}} \sum_{j \in \mathcal{N}} c_{ij} x_{ijk}$$

such that

$$\sum_{k \in \mathcal{V}} \sum_{j \in \mathcal{N}} x_{ijk} = 1 \quad \forall i \in \mathcal{C},$$

$$\sum_{i \in \mathcal{C}} d_i \sum_{j \in \mathcal{N}} x_{ijk} \leq q \quad \forall k \in \mathcal{N},$$

$$\sum_{j \in \mathcal{N}} x_{0jk} = 1 \quad \forall k \in \mathcal{V},$$

$$\sum_{i \in \mathcal{N}} x_{ihk} - \sum_{j \in \mathcal{N}} x_{hjk} = 0 \quad \forall h \in \mathcal{C}, \ \forall k \in \mathcal{V},$$

$$\sum_{i \in \mathcal{N}} x_{i,n+1,k} = 1 \quad \forall f \in \mathcal{V},$$

$$x_{ijk}(s_{ik} + t_{ij} - s + jk) \leq 0 \quad \forall i, j \in \mathcal{N}, \ \forall k \in \mathcal{V},$$

$$a_i \leq s_{ik} \leq b_i \quad \forall i \in \mathcal{N}, \forall k \in \mathcal{V},$$

$$x_{ijk} \in \{0, 1\} \quad \forall i, j \in \mathcal{N}, \ \forall k \in \mathcal{V}.$$

For some discussion regarding VRPTW, we refer to Kallehauge et al. (2005).

## B    Remarks of CG iteration in CSP and VRPTW

### CSP

The CG process of CSP is very similar, as given in Section 2.1, with minor modifications. We take $b = (d_1, d_2, \cdots d_n)$, and in solving (PP), add constraint, $\sum_{j=1}^{n} w_j A_{ji} \leq L$ with the assumtion that $A_{ji}$ belogns to non negative integer.

### VRPTW

Due to its complexity, describing the CG procedure of VRPTW in detail is out of the scope of this paper. However, we remark that we utilize an open source code [3] an experiment done in **RLCG**, which is motivated by the methods provided in Desrochers et al. (1992); Ioannou et al. (2001); Chabrier (2006); El-Sherbeny (2010).

## C    Remarks on LP to ILP

The simplest method to get ILP from LP involves solving the LP relaxation of the RMP and PP and rounding up the results to the nearest integers. However, this method may not always be feasible. Other methods include branch-and-bound, branch-and-price, cutting planes, and metaheuristics. Details of these methods are beyond the scope of this work; therefore, we refer to Wolsey (2020); Schrijver (1998); Desaulniers et al. (2006); Gendreau et al. (2010) for more comprehensive discussions.

## D    Hyperparameters in training RLCG

In this section, we provide hyperparameters in training **RLCG**. For the CSP task, we set the learning rate to be $3e - 4$, the hidden dimension of the GNN model is 32, step penalty of $p = 10$, epoch size $E = 5$, $\alpha = 100$, buffer size 2000 and $\gamma = 0.999$. For the VRPTW task, we set the learning rate to be $1e - 3$, the hidden dimension of the GNN model is 32, step penalty of $p = 10$, epoch size $E = 5$, buffer size 20000, $\alpha = 0.5$ and $\gamma = 0.99$.

---

[3]https://github.com/SimoneRichetti/VRPTW-Column-Generation/

# E   Additional information about the dataset in the training and testing process

We adopt the procedure outlined in Chi et al. (2022). However, for the sake of completeness, we provide the details. In the CSP task, we use 160 instances for the length of the roll 50, 160 instances for the length of the roll 100, and 120 instances for the length of the roll 120. As suggested in Chi et al. (2022), we train both **PPO-CG** and **RLCG** from the easy problems (roll length = 50) to the hard problems (roll length = 200).

For the VRPTW task, we use Solomon (1987) with six differnt types of problems, i.e., C1, C2, R1, R2, RC1, RC2. For the training, C1, R1, and RC1 types are used with different sizes of customers ranging from six to eight. In total, there are 240 instances. Due to the time constraints, only 213 instances are used. We refer to Appendix N for more details on the instances.

# F   Details on the training process

## F.1   Training in CSP tasks

We train our model, **PPO-CG**, with 439 instances in 66.79 **hours**, whereas with the same instances, **RLCG** takes 319.86 **hours**. For the comparison in training time for each instance, see Figure 12. We also provide statistical information about the number of iterations in the training process. See Table 2.

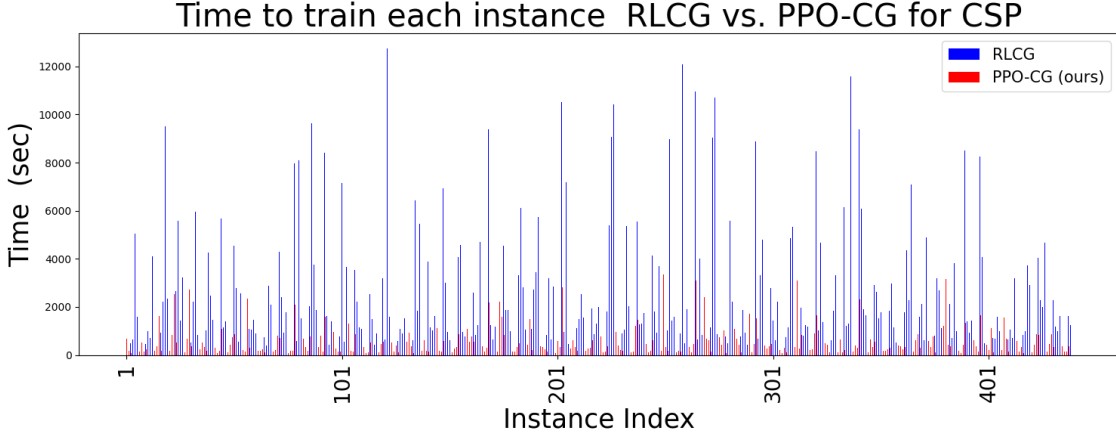

Figure 12: Training time for each instance in CSP

Table 2: Statisctic information about the number of steps in training CSP

| Method | Problem size | $\mu$ | $\sigma$ |
|---|---|---|---|
| | 50 | 37.54 | 8.25 |
| **PPO-CG** | 100 | 68.14 | 14.01 |
| | 200 | 131.06 | 35.7 |
| | 50 | 36.3 | 8.47 |
| **RLCG** | 100 | 67.38 | 16.57 |
| | 200 | 133.04 | 41.46 |

Note that the average number of steps does not show a dramatic change. However, the time to train each model has a dramatic difference since **RLCG** updates its model parameter every step (after Equation (PP is solved, whereas in **PPO-CG**, it updates the parameter after the instance is solved. Therefore **PPO-CG** takes much less time for training.

## F.2    Training in VRPTW tasks

Our model, **PPO-CG**, trains 219 instances in 16.70 **hours**, whereas RLCG trains the same instance in 47.88 **hours**. Please refer to Figure 13 for details. In Table 3, we also provide statistical information about the number of steps taken in the training process.

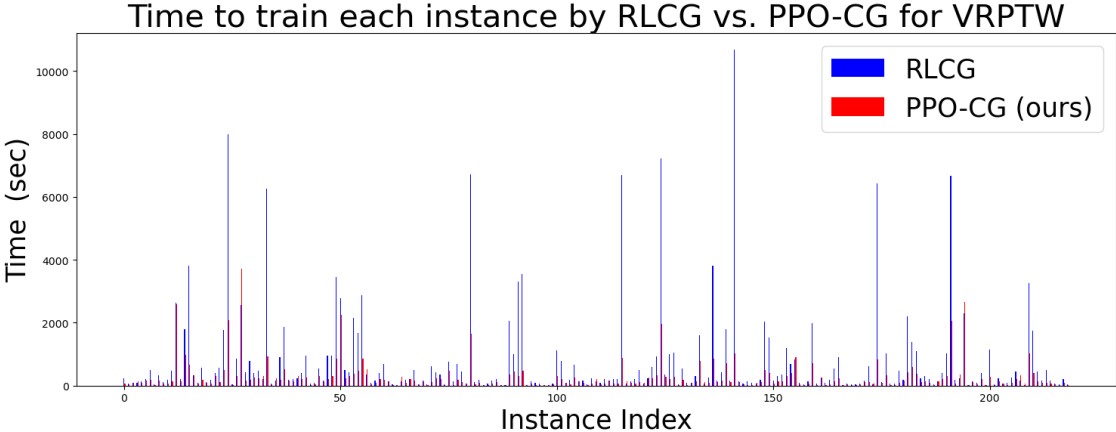

Figure 13: Training time for each instance

Table 3: Statistic information about the number of steps in training VRPTW

| Method | $\mu$ | $\sigma$ |
|---|---|---|
| **PPO-CG** | 29.33 | 17.89 |
| **RLCG** | 49.49 | 28.82 |

We observe that **PPO-CG** has fewer average iteration steps.

## G    Comparison between generating a single trajectory and multiple trajectories in VRPTW task

Training each instance with only one trajectory takes 16.70 **hours**. On the other hand, generating a new trajectory every time each trajectory ends, the training time extends to 47.22 **hours**. We refer to Figure 17 for performance comparison. There are, in total, 37 instances, and as in the comparison presented in Section 42, we exclude one instance with extreme results for clarity. The objective function is identical, and for the execution time and CG iteration number, **PPO-CG** with multiple trajectories shows better results. This is an expected result, given that training with multiple trajectories takes almost three times as much time as training with one trajectory. However, we believe that improvement in training with the new trajectory is minimal compared to the training costs except for a few extreme cases.

## H    Ablation study

This section analyzed the impact of solving PP and updating parameters on learning time. To this end, we train **RLCG** after each training instance. Given that the number of epochs is 5 for **RLCG** and 20

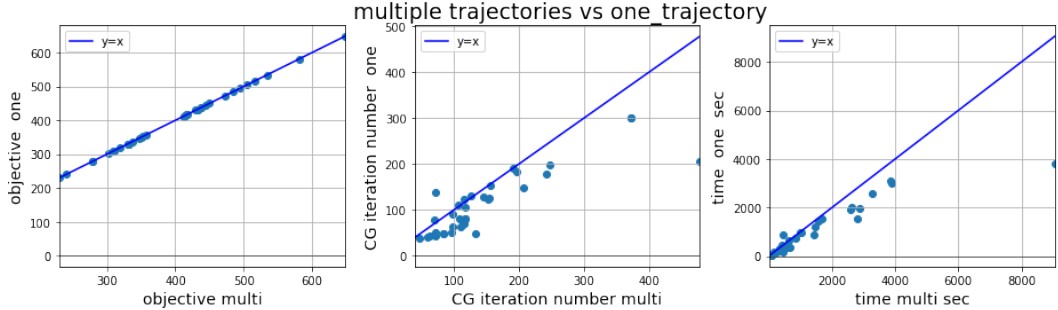

Figure 14: Comparison between **PPO-CG** trained with one trajectory and multiple trajectories in the VPRTW task.

for **PPO-CG**, we trained two models with different epoch sizes. We named each model **RLCG-E5** and **RLCG-E20** for distinction.

### H.1 Comparison in training time and number steps

This subsection reports training time and iteration steps taken during the training process.

Table 4: Overall training time for other algorithm

| Task | Methods | training time(hours) |
|---|---|---|
| CSP | **RLCG** | 319.9 |
| | **RLCG-E5** | 2.8 |
| | **RLCG-E20** | 7.0 |
| | **PPO-CG** (ours) | 66.8 |

Table 5: Average training time and iteration step numbers in training CSP for different models

| Method | Problem size | average iteration steps | average training time(secs) |
|---|---|---|---|
| **PPO-CG** | 50 | 37.5 | 174.5 |
| | 100 | 68.1 | 400.1 |
| | 200 | 131.1 | 1247.9 |
| **RLCG** | 50 | 36.3 | 1010.9 |
| | 100 | 67.4 | 1853.6 |
| | 200 | 133.0 | 5825.1 |
| **RLCG-E5** | 50 | 39.1 | 10.5 |
| | 100 | 68.8 | 17.1 |
| | 200 | 133.1 | 48.1 |
| **RLCG-E20** | 50 | 36.1 | 34.3 |
| | 100 | 68.8 | 49.8 |
| | 200 | 133.0 | 97.4 |

In Figure 15, we present training time for each training instance without including **PPO-CG** and **RLCG**. In Figure16, we include all models.

### H.2 Additional analysis on PP vs parameter updates

To analyze the time taken for learning by solving PP and updating parameters, we assume that the average time spent on PP and parameter updates is the same for **RLCG** and **RLCG-E5** for same prblem size.

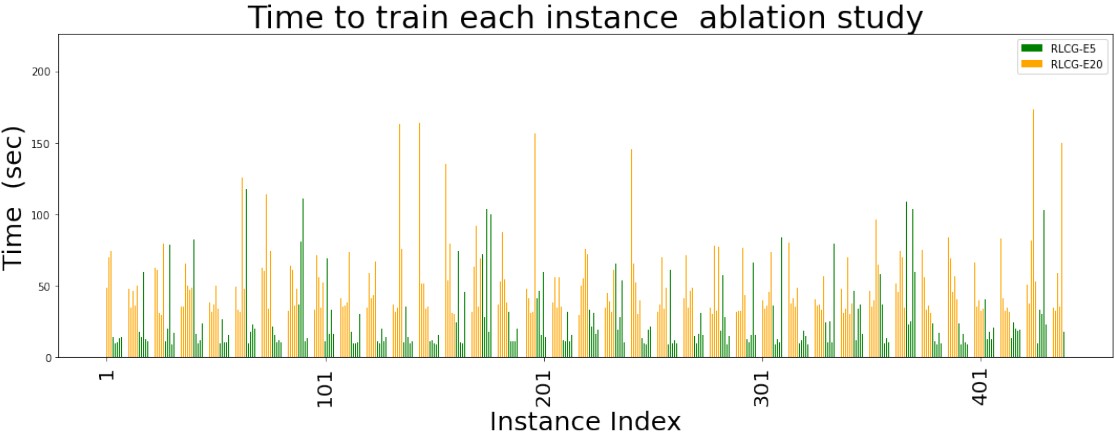

Figure 15: Training time for each instance for **RLCG-E5** and **RLCG-E20**

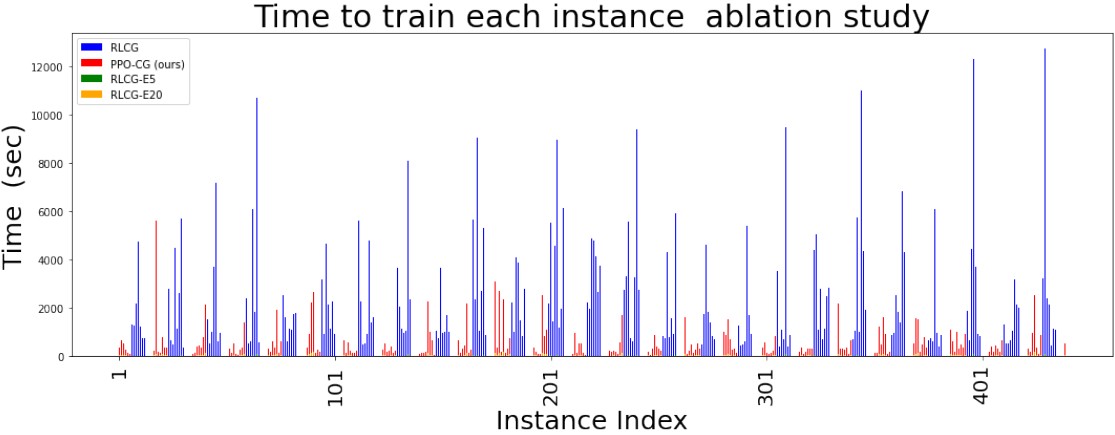

Figure 16: Training time for each instance including all models

Let $E$ be the number of epochs, and we denote $s_{0,i}$ and $s_{1,i}$ for the average step to train problem of size $i$ for **RLCG** and **RLCG-E5**. Let $P_i$ and $U_i$ be the average time it takes to solve (PP) and update parameters for train instance of size $i$, and $T_{0,i}$ $T_{1,i}$ denotes the total time it takes to train each problem of size $i$. We then have the following equations:

$$T_{0,i} = Es_{0,i}(U_i + P_i),$$
$$T_{1,i} = E(U_i + s_{1,i}P_i).$$

Then, from a simple calculation, we have that

$$U_i = \left(\frac{s_{1,i}}{s_{1,i} - 1}\right)\left(\frac{T_{0,i}}{Es_{0,i}} - \frac{T_{1,i}}{Es_{1,i}}\right) \quad \text{and} \quad P_i = \frac{T_{0,i}}{Es_{0,i}} - U_i.$$

From this calculation, we summarize in Table 6 the proportion of time spent on solving PP and parameter updates in the training process.

### H.3   Test results

Finally, we briefly provide test results on **PPO-CG**, **RLCG**, **RLCG-E5** and **RLCG-E20**. Showing that training frequency has an impact on the performance of the model. Also note that there are not many

Table 6: Table showing the proportion of time each element during learning in relation to the total.

| Method | Problem size | solve PP (%) | update parameters(%) |
|--------|--------------|--------------|----------------------|
|        | 50           | 0.7          | 99.3                 |
| **RLCG** | 100        | 1.1          | 98.8                 |
|        | 200          | 11.1         | 88.9                 |

differences in the performance of **RLCG-E5** and **RLCG-E20**, implying that model parameters must be frequently updated to train DQN models properly. Moreover, as stated in Section 4.1, RL-based models do not perform well on CSP problems with $n = 200$, so that in **RLCG-E5** and **RLCG-E20** can have better result.

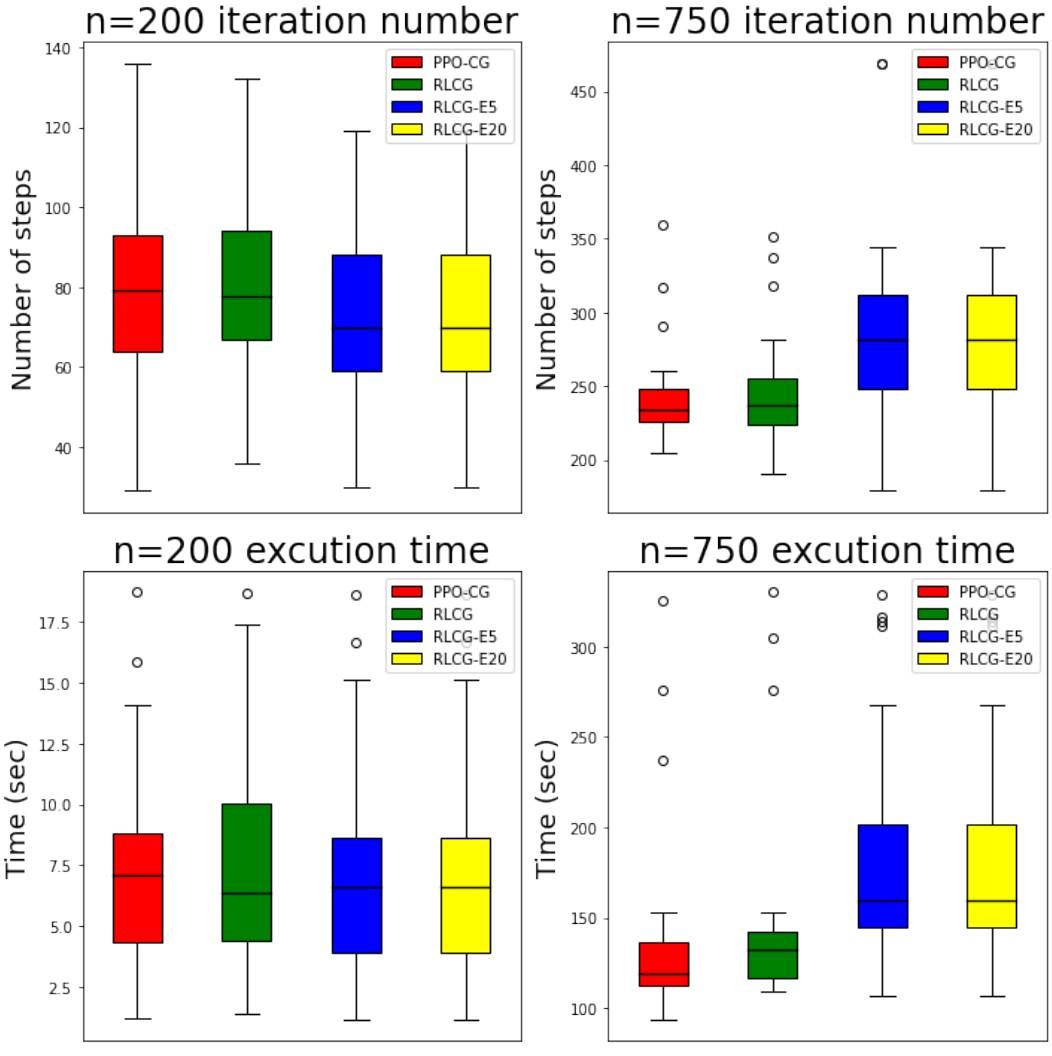

Figure 17: Comparison between **PPO-CG**, **RLCG**, **RLCG-E5** and **RLCG-E20**.

## I   Comparison with the Greedy algorithm

In this ablation study, we compare each algorithm with the greedy algorithm on the test data. While running the greedy algorithm, we compare the action chosen by the greedy algorithm and other RL-based algorithms at each step to see how similar they are. This demonstrates that the DQN algorithm is very similar to the Greedy algorithm. The results are summarized in Table 7 and Table 8.

Table 7: Comparison with the Greedy algorithm with other RL-based algorithms on CSP problems. The third column represents the average percentage of times the action chosen by the RL model matches the action chosen by the greedy algorithm. The last column represents the point at which the action chosen by the RL algorithm first differs from that chosen by the greedy algorithm, normalized for each instance. This normalization accounts for the differing number of steps in each instance. For example, a value of 0.1 indicates that in a sequence of 200 steps, the RL model and the greedy algorithm choose different actions at the 20-th step.

| Method | Problem size | compare with the Greedy (%) | first state is diverges (%) |
|---|---|---|---|
| **RLCG** | 200 | 90.39% | 13.14% |
|  | 750 | 95.29% | 70.29% |
| **RLCG-E5** | 200 | 99.99% | 99.09% |
|  | 750 | 100.00% | 100.00% |
| **RLCG-E20** | 200 | 98.99 % | 98.12% |
|  | 750 | 100.00% | 100.00% |
| **PPO-CG** | 200 | 10.96% | 0.04% |
|  | 750 | 13.02% | 0.09% |

Table 8: Comparison with the Greedy algorithm in VRPTW task. For the explanation on each column, we refer to the caption in Table 7.

| Method | compare with the Greedy (%) | first state is diverges (%) |
|---|---|---|
| **RLCG** | 99.13% | 90.13% |
| **PPO-CG** | 0.05% | 4.19% |

From the results, we can conclude that **RLCG-E5** and **RLCG-E20** are almost identical to the greedy algorithm. **RLCG** also shows little difference from the greedy algorithm in cases involving larger problem sizes, such as CSP with $n = 750$ and VRPTW.

## J   t-SNE analysis for VRPTW tasks

In this section, we aim to understand the exploration effects of each algorithm. To do this, we use t-SNE. We use feature vectors to represent each state. Candidate states are marked in gray during each step for fixed test instances, while the states actually visited are colored. Additionally, the visited states are denoted in darker colors as they are visited later. The image can be found in Figure 18.

## K   Excluded test instance for VPTTW

As mentioned in Section 4.2, we exclude one instance since it exhibits too much time. We summarize in Table 9.

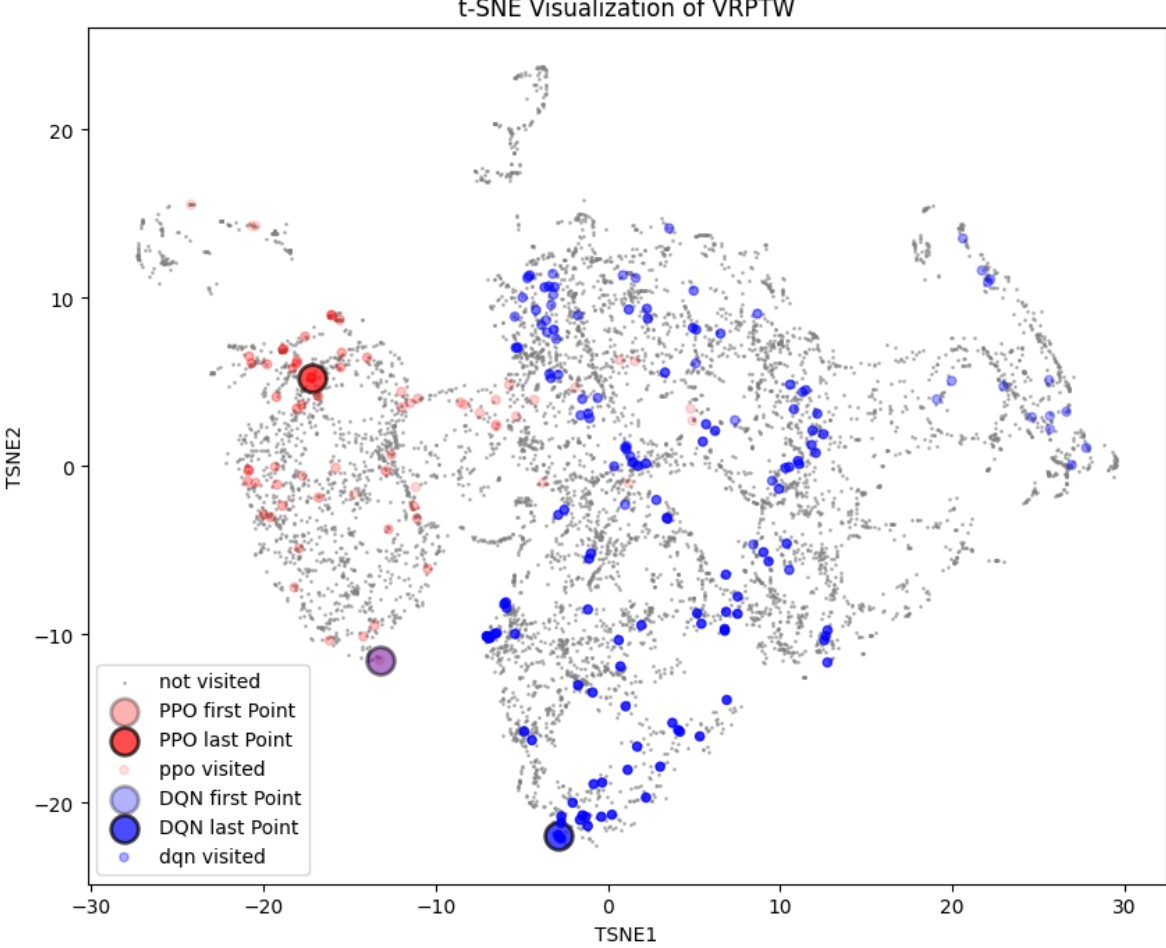

Figure 18: This t-SNE image shows that the PPO algorithm converges with fewer trials, indicating that it has better exploration capabilities for VRPTW tasks.

Table 9: Excluded test instance

| Methods | Time (hours) | CG iteration numbers |
|---------|--------------|----------------------|
| RLCG    | 5.5          | 153                  |
| PPO-CG  | 7.3          | 217                  |
| Greedy  | 5.5          | 153                  |

## L   Node features on CSP and VRPTW tasks

We use node features to encode information about RMP to the graph. as in Gasse et al. (2019); Chi et al. (2022). In the CSP task, we set 9 node features for the column nodes and 2 node features for the constraint nodes. For the column node features, we use

1. reduced cost for each node,

2. number of connected nodes,

3. solution value of RMP corresponding to each column,

4. remaining length of a roll for each pattern,

5. number of iterations that each column node stays in the basis,

6. number of iterations that each column stays out of the basis,

7. if the column left the basis on the last iteration or not,

8. if the column entered the basis on the last iteration or not,

9. action node or not.

For the constraint node features, we use

1. dual value or shadow price of (PP),

2. the number of connected nodes.

The node features for the VRPTW task are very similar. For the column node features, the 4-th and 9-th node features are removed, and the routing cost is added. The constraint node features are the same as CSP.

## M   Details on convergence analysis in Section 4.3

Suppose we have $n$ test instances, with each instance $i$ containing a list of objective values represented as $v_i = [v_{i,1}, \ldots, v_{i,m_i}]$, where $i = 1, \ldots, n$, and $m_i$ is the number of objective values for the $i$-th instance. Let $m = \max_i m_i$. Given that both CSP and VRPTW are minimization problems, we normalize each list of objective values $v_i$ as follows:
$$\tilde{v}_i = \frac{\max_k v_{i,k} - v_i}{\max_k v_{i,k}},$$
where $\max_k v_{i,k}$ is the maximum value in each list, ensuring that $0 \leq \tilde{v}_i \leq 1$, with 1 representing the optimal normalized value.

To standardize the data for analysis, we extend the normalized values $\tilde{v}_i$ to a length of $m$ by appending the value 1 to the end of each list until it reaches the maximum length. This adjustment allows us to compute the average and standard deviation of $\{\tilde{v}_{i,k}\}_{i=1,\ldots,n}$ for each $k = 1, \ldots, m$. These statistics are then used to draw Figure 11, illustrating the convergence behavior of the algorithms across all test instances.

## N   Remark on training and testing result per instances

For more details, we added supplementary files in the output folder of the GitHub repository.

