# OpenReview forum: "Policy optimization in reinforcement  learning for column generation"
_TMLR — Rejected by TMLR_

### Review · Reviewer_M8fT · 2024-05-01

**Summary Of Contributions:**

While column generation is essential for addressing large-scale linear integer programming problems in many industrial domains, few machine learning and reinforcement learning methods are available that enhance the existing column generation algorithm. This paper introduces a new policy optimization RL framework to improve the existing DQN-based column generation framework.

**Audience:**

Yes

**Claims And Evidence:**

Yes

**Requested Changes:**

The authors should consider stating theoretical guarantees for the policy $\pi$ output by Algorithm 1.

**Strengths And Weaknesses:**

Strength: Their method requires 20% of the training time observed with the DQN-based method.
Weakness: lack of theory.

---

> ### Author Response · Authors · 2024-05-30
> **Response to  the commeny by Reviewer M8fT**
>
> We want to thank Reviewer M8fT for the valuable feedback.
>
> These are changes made from the comments.
>
> We added reference on the existing theoretical results on the convergence of PPO and another theoretical framework in the beginning of Section 2.2

---

> > ### Comment · Reviewer_M8fT · 2024-06-23
> > **Thank you for your response**
> >
> > My previous comments were addressed. I don't have further comments.

---

### Review · Reviewer_FJ2H · 2024-05-14

**Summary Of Contributions:**

The paper introduces Proximal Policy Optimization (PPO) to Column Generation (CG) to improve training time. Key contributions include:

- Significant reduction in training time:
  - Merely 20% of the training time in Cutting Stock Problems (CSP).
  - 35% in Vehicle Routing Problems with Time Windows (VRPTW).

**Audience:**

Yes

**Broader Impact Concerns:**

No specific concerns, but the anonymous repository expired during the review period. It is better to check and guarantee the availability of the information with the code release statement.

**Claims And Evidence:**

No

**Requested Changes:**

- Justification for PPO: Provide a clearer explanation of why PPO was chosen over other DRL methods. Discuss its advantages and potential drawbacks in the context of CG.
- Citation Format: Correct the citation format issues throughout the paper.
- Terminology and Concept Accuracy: Revise the discussion on vanilla DQN and the statement about PPO to accurately reflect their characteristics and intended use cases.
- Algorithm Details: Include the missing details on the training of the critic network in Algorithm 1.
- Terminology Consistency: Use standard RL terminology, such as "discount factor," instead of non-standard terms like "reward decay exponent."
- Training Time Comparison: Provide a fair comparison of training times by adjusting the update frequencies and considering the impact of target network update frequencies in DQN.
- Additional Data on PP: Include data on the time spent solving the Pricing Problem (PP) to support the claims of reduced training time.
- Consider Other DRL Methods: If the authors want to provide more evidence about the policy network's potential benefits (not necessary, I know it is costly), it is good to discuss the potential benefits of other DRL methods like IMPALA, SAC, and TD3 for speeding up training and consider their inclusion in the comparative analysis. Additionally, I suggest looking into GDI, as the diversity of data during training could be an implicit factor for efficiency improvement. PPO's sample policy provides an exploration advantage over DQN.
- References:
  - DDPG: Continuous control with deep reinforcement learning. (ICLR 2016)
  - IMPALA: IMPALA: Scalable Distributed Deep-RL with Importance Weighted Actor-Learner Architectures. (ICML 2018)
vSAC: Soft Actor-Critic: Off-Policy Maximum Entropy Deep Reinforcement Learning with a Stochastic Actor. (ICML 2018)
  - TD3: Addressing Function Approximation Error in Actor-Critic Methods. (ICML 2018)
  - GDI: Generalized Data Distribution Iteration. (ICML 2022)

**Strengths And Weaknesses:**

# Strengths:

- Good Attempt: The use of PPO combined with Graph Neural Networks (GNN) in CG is a good attempt for problem-solving.
- Efficiency: Demonstrated significant reductions in training time for CSP and VRPTW.

# Weaknesses:

- Motivation for PPO: The paper does not provide a clear justification for choosing PPO over other DRL methods. PPO is not necessarily superior for CG problems, and the rationale behind its selection should be better articulated. To my knowledge, the main reason why PPO is so popular is that it is simple and easy for implementation and has decent performance, not for its state-of-the-art strength or efficiency.
- Citation Issues: There are several citation format issues that need correction (e.g., replacing citep with citet where appropriate and using the proper publication source rather than arXiv if available).
- Terminology and Concept Accuracy: The description of vanilla DQN's limitations with continuous action space is incorrect (not "tends to underperform" as the cited reference DDPG). Additionally, PPO is not designed to address the specific limitations discussed but is rather an easy-to-implement alternative to TRPO.
- Missing Details: Algorithm 1 seems to lack details on the training of the critic network.
- Terminology Consistency: The term "reward decay exponent" should be replaced with the more commonly used term "discount factor" in RL literature.
- Training Time Comparison: The comparison of training time should consider adjusting the DQN update frequency to match that of PPO. Simply comparing raw training times without accounting for differences in update frequencies and target network updates can be misleading.
- Additional Data: Since the focus is on reducing training time, providing data on the time spent solving the Pricing Problem (PP) would be beneficial.
- Node Selection Contribution: The claim of a novel method for node selection is not sufficiently substantiated, given that RL with GNNs and PPO using GNNs have been previously explored.

---

> ### Author Response · Authors · 2024-05-30
> **Response to Reviewer FJ2H**
>
> We want to thank Review FJ2H for the valuable comments:
>
> These are changes made from the comments:
>
> Justification for PPO: Added paragraph in Section 3.2 and drawbacks are stated in the conclusion section
>
> Citation Format: Made changes
>
> Terminology and Concept Accuracy: The incorrect explanation was removed.
>
> Algorithm Details: Made change in Algorithm 1
>
> removed node selection contribution statement
> Terminology Consistency: we made a change
>
> Training Time Comparison: We added an ablation study in Appendix G
>
> Additional Data on PP: Also added in Appendix G
>
> Consider Other DRL Methods: We briefly mentioned this in the conclusion section since considering other DRL methods costs too much.
>
> Expired github code is renewed

---

> > ### Comment · Reviewer_FJ2H · 2024-06-09
> >
> > * The motivation for adopting PPO instead of DQN for CG is clearer now. I suggest redesigning the experiments to support these new findings and insights (see the following reasons).
> > * The improvement on CSP to RLCG seems to only be in terms of shorter training time. I found that the claim of a 20% training time reduction for CSP might be an overstatement based on the new ablation results. It is evident that the sample efficiency of PPO-CG and RLCG is nearly the same on CSP (according to iteration count). These RL methods do not show clear benefits on smaller problems (750 starts to show some advantage). Furthermore, with RLCG-E5 or RLCG-E20, it is clear that they can achieve much shorter training times with the same iteration count. Thus, the experiments on CSP do not provide clear evidence to support the claim and motivation.
> > * For VRPTW, it seems that PPO-CG can provide some advantage, but the reason is not well explored. I suspect this may be due to better exploration than RLCG, leading to more diverse and useful trials and better sample efficiency. The reason I suggest reading the GDI paper is that it is one of the state-of-the-art DRL algorithms on Atari57 and provides an analysis of why exploration or diversity is often the reason for better performance. Although it is not specifically about PPO, it serves as evidence to argue that the improvement might stem from a better exploration strategy compared to DQN (with epsilon-greedy).
> > * I think it is interesting to improve general constrained optimization problems with RL techniques. This paper's focus on CG problems has its value; however, in its current form, I believe there is still much room for supporting the motivation and addressing the specific problem. Otherwise, it appears as just another engineering attempt to apply PPO to CG instead of DQN, reporting less training time, which is not surprising to an ML audience familiar with RL. The audience of TMLR is likely more interested in "Why should we use PPO for CG?" This should have a clear reason (your new motivation part gives some insights) and provide the corresponding experiments to support it.

---

> > > ### Author Response · Authors · 2024-06-11
> > > **Reply for the responseto the comment by Reviewer FJ2H**
> > >
> > > We would like to appreciate a detailed review after the revision.
> > >
> > >
> > > *"These RL methods do not show clear benefits on smaller problems (750 starts to show some advantage). Furthermore, with RLCG-E5 or RLCG-E20, it is clear that they can achieve much shorter training times with the same iteration count. Thus, the experiments on CSP do not provide clear evidence to support the claim and motivation."
> > > -> First, for smaller problems, RL methods (PPO-CG and RLCG) do not perform better. As stated in Section 4.2, the Greedy method or Naive CG method is good enough. However, as stated, RL methods work better on larger problems such as CSP 750 or VRPTW.
> > > -> Second, Figure 17 can be misleading because RLCG_E5 and RLCG_E20 work better on CSP. However, both RLCG_E5 and RLCG_E20 behave just like the Greedy algorithm, and showing better performance indicates that they are undertrained or did not converge to the optimal solution, possibly to a suboptimal solution. We are running additional experiments to show this in detail. So, showing better performance on CSP 200 problems for RLCG_E5 and RLCG_E20 is other proof that more interaction is required to train DQN.
> > >
> > > *"For VRPTW, it seems that PPO-CG can provide some advantage, but the reason is not well explored. I suspect this may be due to better exploration than RLCG, leading to more diverse and useful trials and better sample efficiency. The reason I suggest reading the GDI paper is that it is one of the state-of-the-art DRL algorithms on Atari57 and provides an analysis of why exploration or diversity is often the reason for better performance. Although it is not specifically about PPO, it serves as evidence to argue that the improvement might stem from a better exploration strategy compared to DQN (with epsilon-greedy)."
> > > -> We agree that better performance of PPO-CG on VRPTW is due to better exploration than RLCG. As stated in GDI paper, interaction in VRPTW is very limited because we need to calculate what is available route, which is computationally very heavy. We are in the process of adding an explanation on this issue.
> > >
> > > Thank you again for the advice.

---

> > > > ### Author Response · Authors · 2024-06-12
> > > > **Additional result**
> > > >
> > > > Please refer to Appendix I and Appendix J for the revised manuscript.
> > > >
> > > > Thank you for your attention in this manuscript.

---

> > > > > ### Comment · Reviewer_FJ2H · 2024-06-18
> > > > >
> > > > > Do you consider trying different $\epsilon$-greedy scheduling for DQN and different entropy regularization for PPO (I am not sure if your PPO implementation includes entropy regularization) to formally discuss the exploration problem in CG?
> > > > >
> > > > > Actually, I think the main value of this paper would be answering this problem: whether we need a better exploration strategy in CG with RL, and if RL methods with an explicit policy fit this need.
> > > > >
> > > > > Thus, this should be the main topic of the paper rather than simply reporting less system running time than DQN for a specific CG problem. So, I think you need to reorganize this paper rather than provide some side evidence in the appendix.
> > > > >
> > > > > If you want to further support the exploration idea, try adding some redundant actions to the problem that significantly influence the exploration property.
> > > > >
> > > > > Besides, you mention that expert heuristics or greedy methods can do well in small problems.
> > > > > What evaluation function or value is used in the greedy or heuristic methods? It should be described more in the main paper, especially if RL methods cannot beat them.

---

> > > > > > ### Author Response · Authors · 2024-06-19
> > > > > > **To Reviewer FJ2H**
> > > > > >
> > > > > > Thank you for your great interest in this study and for pointing out valuable insights. Reorganizing this paper will take some time, so we kindly ask for your patience.

---

### Review · Reviewer_q9PB · 2024-05-16

**Summary Of Contributions:**

The paper looks at the problem of Column generation (CG) for addressing large-scale linear integer programming problems with the use of reinforcement learning. This paper introduces a new policy optimization RL framework to improve the existing DQN-based CG approach, particularly training time. The work "is motivated by the methods provided in (Morabit et al., 2021) and (Chi et al., 2022)". The DQN-based approach in (Chi et al., 2022) is called RLCG, and their method is named PPO-CG.

**Audience:**

Yes

**Claims And Evidence:**

Yes

**Requested Changes:**

Scientific clarity can be improved
- What does the following mean in 4.3 with respect to standard deviation? "The objective numbers are scaled from 0 to 1, and we added ±1 standard deviation."
- In Fig 12, what do the dots mean? How can we see whether the results are significant? (From that plot, we can't).

Suggestions for better writing/presentation clarity:
- "Although our problem focuses on Integer Linear Programming, we shall provide a CG method for linear programming in this subsection". Why? At the end of the section, it is written "The column generation process for ILP is very similar, with some modiﬁcations for each task." but I believe this comes a bit late and is not fully clear on the part about "some modiﬁcations for each task". Citing some relevant work might for instance help if the theory would be too long to cover.
- Small remark: In Equation 3, it is a bit confusing to have $A_t^1$ and $A_t^2$ that are not the advantage function while $A_t$ is used as the advantage function.
- Figure 7, 8, 11 are not directly visually intuitive (e.g. "y=x" and what the dots mean). The caption does not provide any explanation.
- You should only show the significant digits in numbers, e.g. we definitely do not need the digits after the comma in "For the extreme case is when PPO-CG takes 26200.88 seconds to solve with 217 iterations, RLCG takes 20138.50 seconds with 153 iterations". In addition providing these results in plain text instead of through tables or plots makes the paper less easily readable.
- Figure 12: typo in the title
- Caption often lack accurate description, e.g. "Figure 9: Bar chart for four methods for n = 200 and n = 750 in CSP)" does not provide information about the fact that two of the bar charts are about execution time and two about "number of steps".
- (not a major issue but could strengthen the paper) The overall writing style has room for improvements.

**Strengths And Weaknesses:**

Strengths
- The paper tackles an interesting problem and shows a major reduction in execution time as compared to a baseline

Weaknesses
- The paper has areas of improvements concerning the writing and presentation (see request for changes).
- Overall, the results are in different cases not convincingly discussed. For instance, it is not fully clear why the number of steps are comparable but the training time is not: it takes "66.79 hours, whereas with the same instances, RLCG takes 319.86 hours". In general, the paper uses sentences such as "To our surprise, this method has shorter training time (...)". It would be interesting to have more insights on why.

---

> ### Author Response · Authors · 2024-05-30
> **Response to Reviewer q9PB**
>
> We would like to thank Reviewer q9PB for the valuable feedback.
>
> These are changes made from the comments.
>
> What does the following mean in 4.3 with respect to standard deviation? : Added explanation in Appendix K`
>
> We added detailed captions for the Figures
>
> We explained how to solve integer linear programs from linear programs in Section 2.1 and Appendix C
>
> Explanation of PPO algorithm has changed

---

### Author Response · Authors · 2024-05-30
**Response to reviewes: Additional appendix**

We added addtional ablation studies in Appendix I and Appendix J.

---

### Decision · Action_Editor_25sz · 2024-07-06

**Recommendation:** Reject

**Comment:**

All reviewers acknowledged the potential merit of the proposed approach regarding improved efficiency.

However, there are many concerns regarding, lacking of motivation for PPO, clarity, fairness of comparisons, unconvincing discussion of results, and doubts about the results w.r.t. solution quality.

The requested revision of the paper would lead to a substantially different version of the current submission.

**Audience:**

At this point, the findings are not sufficiently supported by the motivation and experimental results.

**Claims And Evidence:**

The clarity and motivation are criticized by most reviewers. Furthermore, the experimental results do not fully support the claims.